# Inhibitory Effect of Styrylpyrone Extract of *Phellinus linteus* on Hepatic Steatosis in HepG2 Cells

**DOI:** 10.3390/ijms24043672

**Published:** 2023-02-12

**Authors:** Chun-Hung Chiu, Ming-Yao Chen, Jun-Jie Lieu, Chin-Chu Chen, Chun-Chao Chang, Charng-Cherng Chyau, Robert Y. Peng

**Affiliations:** 1Research Institute of Biotechnology, Hungkuang University, Shalu District, Taichung City 43302, Taiwan; 2Department of Program in Animal Healthcare, Hungkuang University, Shalu District, Taichung City 43302, Taiwan; 3Division of Gastroenterology and Hepatology, Department of Internal Medicine, School of Medicine, College of Medicine, Taipei Medical University, Taipei 11031, Taiwan; 4Division of Gastroenterology and Hepatology, Department of Internal Medicine, Taipei Medical University—Shuang-Ho Hospital, New Taipei City 235041, Taiwan; 5Grape King Biotechnology Center, Longtan District, Taoyuan 325002, Taiwan; 6Division of Gastroenterology and Hepatology, Department of Internal Medicine, Taipei Medical University Hospital, Taipei 11031, Taiwan; 7TMU Research Center for Digestive Medicine, Taipei Medical University, Taipei 110, Taiwan; 8Graduate Institute of Clinical Medicine, College of Medicine, Taipei Medical University, Taipei 110301, Taiwan

**Keywords:** NAFLD, *Phellinus linteus*, hepatic steatosis, HepG2 cells, oil droplet accumulation, SIRT1/AMPK/PGC1-α pathway

## Abstract

The prevalence of nonalcoholic fatty liver disease (NAFLD) is estimated to be approximately about 25.24% of the population worldwide. NAFLD is a complex syndrome and is characterized by a simple benign hepatocyte steatosis to more severe steatohepatitis in the liver pathology. *Phellinus linteus* (PL) is traditionally used as a hepatoprotective supplement. Styrylpyrone-enriched extract (SPEE) obtained from the PL mycelia has been shown to have potential inhibition effects on high-fat- and high-fructose-diet-induced NAFLD. In the continuous study, we aimed to explore the inhibitory effects of SPEE on free fatty acid mixture O/P [oleic acid (OA): palmitic acid (PA); 2:1, molar ratio]-induced lipid accumulation in HepG2 cells. Results showed that SPEE presented the highest free radical scavenging ability on DPPH and ABTS, and reducing power on ferric ions, better than that of partitions obtained from n-hexane, n-butanol and distilled water. In free-fatty-acid-induced lipid accumulation in HepG2 cells, SPEE showed an inhibition effect on O/P-induced lipid accumulation of 27% at a dosage of 500 μg/mL. As compared to the O/P induction group, the antioxidant activities of superoxide dismutase, glutathione peroxidase and catalase were enhanced by 73%, 67% and 35%, respectively, in the SPEE group. In addition, the inflammatory factors (TNF-α, IL-6 and IL-1β) were significantly down-regulated by the SPEE treatment. The expressions of anti-adipogenic genes involved in hepatic lipid metabolism of 5’ adenosine monophosphate (AMP)-activated protein kinase (*AMPK*), sirtuin 1 (*SIRT1*) and peroxisome proliferator-activated receptor gamma coactivator 1-alpha (*PGC-1α*) were enhanced in the SPEE supplemented HepG2 cells. In the protein expression study, p-AMPK, SIRT1 and PGC1-α were significantly increased to 121, 72 and 62%, respectively, after the treatment of SPEE. Conclusively, the styrylpyrone-enriched extract SPEE can ameliorate lipid accumulation and decrease inflammation and oxidative stress through the activation of SIRT1/AMPK/PGC1-α pathways.

## 1. Introduction

The incidence and prevalence of nonalcoholic fatty liver disease (NAFLD) are rapidly rising worldwide [1,2]. NAFLD is a common chronic pathology characterized by serial progressive histological alterations to the hepatic parenchyma [3,4]. The global prevalence of NAFLD is approximately 25.24% [5], ranging from 13% in Africa [6] to 42% in Southeast Asia [7], contrasting with 37.50% in the USA and 29.81% in the mainland China [5]. Currently, NAFLD is already known as the fastest growing cause of hepatocellular carcinoma (HCC) in the USA, France and the UK [8]. Beginning with a simple fat accumulation in hepatocytes without inflammation to a more severe histological picture comprising hepatocyte injury, fibrosis and inflammation; a more advanced pathological stage known as non-alcoholic steatohepatitis (NASH) can be accessed in some cases [9]. Up to 20% of patients with NASH may develop cirrhosis [10]. The incidence of nonalcoholic steatohepatitis (NASH) is projected to increase by up to 56% in the next 10 years [8]. The estimated annual incidence of HCC ranges from 0.5% to 2.6% among patients with NASH cirrhosis, while the incidence of HCC among patients with non-cirrhotic NAFLD is lower—approximately 0.1 to 1.3 per 1000 patient-years [8]. Although the incidence of NAFLD-related HCC is lower than that of HCC of other etiologies such as hepatitis C, more people have NAFLD than other liver diseases [8]. 

*Phellinus linteus* (Berkeley and M. A. Curtis) Teng,—known as ‘Sanghuang’ mushroom in China and Korea, and ‘Meshimakobu’ in Japan—is a popular medicinal mushroom widely used in China, Korea, Japan and other Asian countries [11]. For over 2000 years, *P. linteus* has been recognized as beneficial to health in ancient medicine [12]. *P. linteus* comprises various bioactive components, such as polysaccharides, phenylpropanoids, furans [11], flavonoids [12], triterpenoids, steroids [13,14,15], pyranones and styrylpyrones [16,17,18,19]. 

In traditional Chinese medicine, it has been proved to be an effective therapeutic and nutritional agent [11] used for its hepatoprotective, anti-cancer, anti-inflammation, immune-modulation, antioxidative, or antifungal activities, as well for its antidiabetic and neuroprotective effects [11]. Lee et al. [19] demonstrated a potent lipase inhibitor in a methanol extract of the *Phellinus linteus* fruiting body, which exhibited a lipase-inhibitory activity of 72.8%. In addition to lipase-inhibitory activity, the inhibitor demonstrated a 59.4% superoxide-dismutase-like activity and a 56.3% acetylcholinesterase-inhibitory activity [19]. Previously, we reported that styrylpyrones fractionated from *P. linteus* mycelia effectively alleviated NAFLD in a mice model [20]. In this, hispidin and hypholomine B were characterized as the main compounds in the extract using LC/MS analysis [20]. Hispidin has been reported to demonstrate anti-oxidant, cytotoxic, anti-inflammatory, anti-viral and anti-dementia activities [21]. Meanwhile, hypholomine B is reported with anti-oxidant, anti-diabetes and anti-inflammatory activities [21]. In addition, the different species *P. hispidus* (synonym of *Inonotus. hispidus*) from that*P. linteus* has been used for treating cancer, diabetes and stomach problems and has also been indicated to contain rich styrylpyrones [22]. In order to determine whether the HepG2 cell model will respond similarly and provide a more time-saving and cost-saving experimental protocol, we performed this study. 

## 2. Results

### 2.1. Yield of Extraction from Different Solvents and the Antioxidant Activity of Each Extract

As previously reported [20], based on their total polyphenol content, the optimum extractability from the freeze-dried PL mycelia was obtained with 75% methanol, reaching a level of 70.13 ± 5.90 mg/g, which was further partitioned with different solvents including n-hexane (9.24 ± 0.04 mg/g), ethyl acetate (52.58 ± 0.03 mg/g), n-butanol (11.11 ± 0.03 mg/g) and water (1.87 ± 0.04 mg/g). The styrylpyrone-rich extract identified in the ethyl acetate fraction in the previous report [20] was designated as SPEE.

### 2.2. Antioxidant Activity of SPEE 

In all three antioxidant tests, the SPEE partition dose-dependently showed the greatest capability (Figure 1). At 1000 μg/mL, its scavenging capability for DPPH reached 68% (Figure 1A), 88% for ABTS^•+^ free radicals (Figure 1B) and 95% for ferric ion reduction (Figure 1C), comparable to the reference compound BHT.

### 2.3. HPLC Analysis of SPEE

In line with our LC-ESI-MS/MS analysis results [20], the HPLC analysis showed that the SPEE apparently revealed five peaks, which were assigned to hispidin (peak 1), MW 490 unassigned compound (peak 2), hypholomine B (peak 3), hypholomine B isomer (peak 4) and unidentified compound (peak 5), respectively (Figure 2). Hypholomine B was the major constituent; its content prevailed over hispidin. 

### 2.4. Effects of SPEE on O/P-Induced Cytotoxicity in HepG2 Cells 

The cytotoxicity of O/P on the HepG2 cell line became apparent at a dose of 0.4 mM (Figure 3A). the cell viability declined in a dose-dependent manner until 0.8 mM (Figure 3A). In contrast, the cytotoxicity of SPEE was apparently found only at doses larger than 250 μg/mL (Figure 3B). Thus, the HepG2 cells were insulted with 0.7 mM O/P, and the rescue by SPEE was tested at a dose range of 100 to 500 μg/mL (Figure 3C). The results presented in Figure 3C showed that a dose of 0.7 mM O/P suppressed the cell viability to a survival rate of 56%. In the presence of SPEE, the cell viability was retained at 74, 73, and 56%, respectively, at 100, 250, and 500 μg/mL SPEE (*p* < 0.05).

### 2.5. Effects of SPEE on O/P-Induced Lipid Droplet Deposition 

O/P at 0.3 mM induced, significantly, lipid droplet deposition in the HepG2 cell line. Estimation by 2’,7’-dichlorodihydrofluorescein diacetate (DCFH-DA) fluorescence intensity from the Oil Red and Nile red staining revealed that, in the presence of SPEE 100, 250 and 500 μg/mL, the percent lipid droplet deposition was dose-dependently reduced by 11.5, 23 and 27%, respectively (Figure 4).

### 2.6. Effects of SPEE on O/P-Induced Production of Intracellular Reactive Oxygen Species (ROS) 

The intense green fluorescence revealed that severe intracellular ROS stress was induced by O/P 0.3 mM (Figure 5). The intracellular DCFH-DA fluorescence intensity reached 139% compared to the control (100%). SPEE at 250 and 500 μg/mL effectively alleviated the fluorescence intensity and reduced approximately 20% of the ROS intensity (Figure 5). The intracellular level of MDA was completely suppressed, while the levels of intracellular antioxidant enzyme SOD was enhanced by 1.6 and 1.7 fold, catalase was enhanced by 1.32 and 1.35 fold and GPX was stimulated to reach 1.62 and 1.67 fold, respectively (Figure 5).

### 2.7. Effects of SPEE on O/P-Induced Antioxidant Enzyme Activities 

Malondialdehyde (MDA) is the peroxidation product from the oxidation of polyunsaturated fatty acids, and is one of the major markers for lipid peroxidation of cell membrane. In Figure 6A, MDA content of HepG2 cells in the O/P group increased significantly (*p* < 0.01) compared to the control group, while the intervention of SPEE significantly (*p* < 0.001) decreased the MDA content even at the low SPEE dosage of 100 μg/mL, with a similar effect of silibinin at 25 μM, a natural herbal medicine for the treatment of liver disorders. 

The intracellular antioxidant enzymes SOD, CAT and GPx are important components for the protection of cellular functions. These antioxidant enzymes act as antioxidants to scavenge free radicals under oxidative stress [23]. As shown in Figure 6B–D, significantly increased activity in the antioxidant enzymes SOD, CAT and GPx was observed in the SPEE- and the positive-control-treated groups when compared with the O/P group. Overall, the high dose of 500 μg/mL presented equivalent effects on the increasing of antioxidant enzyme activities, except for GPx.

### 2.8. Effects of SPEE on O/P-Induced Pro-Inflammatory Signals 

Lipid accumulation increases the risk of lipid peroxidation and the production of inflammatory cytokines in the liver cells, and is closely associated with the pathogenesis of NAFLD. To test whether SPEE could improve the inflammation induced by O/P damage, we measured the levels of inflammatory markers TNF-α, IL-6 and IL-1β. O/P (0.3 mM) induced the expressions of TNF-α, IL-6 and IL-1β (Figure 7). In the presence of SPEE 250 and 500 μg/mL, the upregulated TNF-α was suppressed by 20 to 25% (Figure 7C. In contrast, SPEE at 100 μg/mL did not show any significant effect. The upregulated IL-6 was effectively suppressed by SPEE by 40%, and that of IL-1β by 34 to 40% (Figure 7B). 

### 2.9. Effects of SPEE on O/P-Induced Expression of Intracellular mRNAs 

O/P downregulated several genes involving SIRT1, PGC1-α and AMPK, and upregulated the NFκB gene (Figure 7A). On the other hand, O/P downregulated proteins Sirt-1, PGC1-α and p-AMPK, and upregulated NFκB, but did not affect that of AMPK (Figure 7B,C).

SPEE at 100, 250 and 500 μg/mL reversed these effects, although 100 μg/mL showed much less of an effect (Figure 7A).

## 3. Discussion 

### 3.1. The Plausible Strong Antioxidant Activities of SPEE

As shown in our previous report, in the primary fractionation with 75% methanol, the polyphenol content was reported to be 70.13 ± 5.90 mg/g, and after being partitioned with ethyl acetate (SPEE), the content was still maintained at 52.58 ± 0.028mg/g [20], the highest among all solvents used for partition. Polyphenols, particularly hispidin and hypholomine B—found to be abundant in the SPEE partition (Figure 2)—are of particular interest because they occur in high concentration in *P. linteus* mycelia. 

The styrylpyrone-type polyphenols of PL, i.e., hypholomine B and hispidin, have been reported to have a significant scavenging activity against radical species in a concentration-dependent manner. In regards to ABTS^•+^ scavenging capabilities, hypholomine B’s was found to be four times stronger than that of Trolox, and superior to that of hispidin [24]. In the study, we found that SPEE showed antioxidant activity comparable to that of BHT, a well-known standard antioxidant (Figure 1).

### 3.2. How Did SPEE Rescue the HepG2 Cell Viability Insulted by O/P?

SPEE contained a tremendous quantity of polyphenols. The most effective protective effect of SPEE in rescuing HepG2 cell line viability from the insult of O/P implicated the overwhelmed antioxidant activity of SPEE against the tremendous production of intracellular ROS (Figure 3, Figure 5). Literature has reported that overloaded palmitate can cause mitochondrial dysfunction and ATP depletion in HepG2 cells, while the antioxidant may restore the mitochondrial respiratory chain activity in the fatty-acid-overloaded hepatocyte [25]. Furthermore, apart from single antioxidant capacity, the major contribution of antioxidants may regulate the cellular antioxidative system through the ERK/Nrf2 pathway [25] in order to regulate the mitochondrial metabolism. In addition, adiponectin receptor 1 (AdipoR1), ubiquitously expressed in cells, has been reported to act through activation of the 5-adenosine-monophosphate-activated protein kinase (AMPK) pathway, which in turn stimulates glucose uptake and decreases fat accumulation [26]. In the transgene AdipoR1 study, overexpression of AdipoR1 attenuates palmitate-induced apoptosis in HepG2 cells [27]. Whether a similar phenomenon could occur in HepG2 cells remains a large space for further investigation.

### 3.3. SPEE Dose-Dependently Inhibited the Intracellular Lipid Droplet Deposition 

Fatty acids play important roles in the progression of a diversity of diseases. The free fatty acids elevated in patients with NASH are positively correlated with the severity of disease [28]. Saturated free fatty acids (FFAs) such as palmitate in the circulation are apt to cause endoplasmic reticulum (ER) stress, apoptosis and insulin-resistance in peripheral tissues [29,30,31]. Figure 4, in reality, was mimicking such a condition (Figure 4). ER stress refers to an imbalance between the demand for protein folding and the protein folding capacity of the ER [29]. Previous studies indicate that the antioxidant resveratrol, a SIRT1 activator, significantly inhibits palmitate-induced ER stress [32]. In the study, we have found that the expressions of both SIRT1 gene and protein were upregulated significantly after the supplementation of SPEE under the insult of O/P (Figure 8). This study suggested that the impaired functions of ER affected by O/P might be reversed through the activation of SIRT1 expression.

### 3.4. SPEE Suppressed ROS Signaling and Induced Antioxidant Enzyme Activities 

The DCFH-DA analysis showed O/P induced production of a significant quantity of ROS signaling (Figure 5) and MDA (Figure 6A). O/P downregulated GPx (Figure 6D), but did not affected SOD and catalase (Figure 6B,C). SPEE dose-dependently enhanced the activations of SOD, catalase and GPx (Figure 6). Evidently, SPEE inhibited ROS signaling through two ways: one via direct scavenging free radicals by hypholomine B and hispidin, and the other via indirect upregulation of the activities of antioxidant enzymes (such as SOD, CAT and GPx) (Figure 6). 

### 3.5. SPEE Alleviated ROS-Induced Pro-Inflammatory Signals under Insult of O/P

O/P via ROS signaling induced the activation of pro-inflammatory signaling pathways, such as TNF-α, IL-6 and IL-1β (Figure 7), and accompanied byinduced injury of hepatic cell (Figure 3A). The major active compound silibinin—found in silymarin, itself extracted from milk thistle fruits—has been demonstrated to possess therapeutic effects for high-fat-induced hepatic steatosis, insulin resistance and glucose metabolism through the inhibition of inflammation levels in mice [33]. TNF-α, IL-6 and IL-1β were higher in the O/P group than in the control group. Conversely, SPEE and silibinin remarkably diminished TNF-α, IL-6 and IL-1β levels (Figure 7).

The IL-1β–TNF-α–IL-6 pro-inflammatory cascade via NFκB and STAT3 pathways was identified as the key driver of inflammation [34]. The non-histone protein targets of SIRT1 are diverse, such as PGC1-α and NF-κB, etc., which are related to the cell apoptosis, oxidative stress and inflammatory response [35]. SPEE alleviated such events in a dose-dependent manner through the upregulation of mRNA and protein expression levels of NFκB, SIRT1 and PGC-1α (Figure 8). 

### 3.6. SPEE Modulated and Alleviated O/P-Induced Intracellular mRNA Expressions

O/P significantly downregulated several genes involving *SIRT1* (*p* < 0.05), *PGC1-α* (*p* < 0.001) and *AMPK* (*p* < 0.001), and upregulated genes such as NFκB (*p* < 0.001) (Figure 7A); the changes among these genes did show apparent improvement afetr the co-treatment with SPEE or silibinin (Figure 8A). Among the protein expressions, Sirt-1, NFκB, PGC1-α and p-AMPK also presented similar trends to that of gene expression in the SPEE-combined O/P treatment group (Figure 8B,C). However, the protein expressions in the silibinin treatment seemed to present a better ameliorating effect than that of SPEE treatment (Figure 8B,C). The differences might be caused by the time difference between drug absorption and reaction [36]. It is worth noting that SPEE may affect the –upregulation of antioxidant enzymes’ activities (SOD and CAT), with the same effects as the silibinin (Figure 6B,C). 

AMPK is a cellular energy sensor that maintains the homeostasis for low ATP levels or high ADP/AMP ratios. AMPK activation positively regulates signaling pathways that replenish cellular ATP supplies, including fatty acid oxidation and autophagy. As a result, it activates pathways that produce ATP through glucose, lipid and mitochondrial metabolism pathways, increasing ATP levels. Conversely, pathways that consume ATP are inhibited by AMPK [37]. 

SIRT1 and AMPK have been shown to play many similar roles in light of their ability to respond to stress and nutrient status [38]. Upon energy-stress stimuli, the AMPK signaling pathway is upregulated, resulting in the increase in intracellular NAD+ levels. At the same time, histone deacetylase SIRT1 and/or transcription factors such as PGC-1α are phosphorylated in an AMPK-dependent manner [38] (Figure 9).

Both AMPK and SIRT1 are common in many aspects, and cooperatively activate PGC1-α, which in turn activates mitochondrial function and fatty acid oxidation (Figure 9). In addition to energy stress, SPEE activated AMPK, SIRT1 and PGC1-α, upregulating antioxidant genes and, at the same time, downregulating the pro-inflammatory genes (Figure 9). In glucose metabolism, glucose uptake and glycolysis are upregulated, while gluconeogenesis and glycogenesis are suppressed [37]. As for lipid metabolism, lipolysis and β-oxidation are enhanced; conversely, lipogenesis and cholesterol biosynthesis are inhibited [37]. Overall, NFκB-induced pro-inflammatory pathways are suppressed, SIRT1, AMPK and PGC1-α pathways are stimulated and ROS signaling is inhibited.

A summarized result is shown in Figure 9. 

## 4. Materials and Methods

### 4.1. Chemicals and Instruments

Methanol, n-hexane, ethyl acetate, n-butanol, petroleum ether, isopropanol, chloroform, dichloromethane, diethyl ether and dimethyl sulfoxide (DMSO) were of reagent-grade and purchased from Merck (Darmstadt, Germany). The 2,2-Diphenyl-1-picrylhydrazyl (DPPH) was obtained from Fluka (Buchs, Switzerland). Acetonitrile (LC/MS-grade), butylated hydroxytoluene (BHT), potassium hexacyanoferrate (K3Fe(CN)6), ropidium iodide (PI), Folin–Ciocalteu reagent, trypan blue, 3-(4,5-Dimethylthiazol-2-yl)-2,5-diphenyltetrazolium bromide (MTT), Oil Red O solution 0.5% in isopropanol, Nile red (BioReagent, suitable for fluorescence), 2′,7′-Dichlorodihydrofluorescein diacetate (DCFH-DA), 2′,2′-azinobis (3-ethyl-benzothiazoline-6-sulfonic acid) (ABTS^•+^), hispidin and bovine serum albumin (BSA) were purchased from Sigma-Aldrich (St. Louis, MO, USA). Dulbecco’s modified Eagle’s medium (DMEM) and trypsin–EDTA solution were provided by Hyclone (Logan, UT, USA).

A series 1260 Infinity HPLC system equipped with a model G1379B degasser, model G1312B binary gradient pump, model G1329B autosampler, model G1316A column oven, model G1315D photodiode array detection (PDA) system and a triple quadruple mass spectrometer 6420A were purchased from Agilent Technologies (Santa Clara, CA, USA). 

### 4.2. Culture of HepG2 Cells 

The HepG2 human hepatocellular carcinoma cell line (ATCC CRL-11997) was purchased from the Bioresources Collection and Research Center (Shin-Chu, Taiwan). HepG2 cells were cultured in a minimum essential medium (MEM) containing 10% fetal bovine serum, 1% penicillin-streptomycin, 1% sodium pyruvate and 1% non-essential amino acids and were maintained in humidified 5% CO_2_/95% air at 37 °C.

### 4.3. Induction of HepG2 Cell Line with O/P

The HepG2 cell line was cultured in minimum essential medium alpha medium (Gibco, Grand Island, NY, USA) and induced with O/P = oleic acid/palmitic acid = 2:1 molar ratio [39]. 

### 4.4. MTT Assay

According to the method of Mosmann et al. [40], the MTT test solution (×10 dilution, first filtered through a 0.22 μm micropore membrane filter and stored at 4 °C. A subsequent 1/10 dilution of this in sterilized water was added to each cell-cultured well and mixed thoroughly. The MTT solution was removed, and 100 μL DMSO was added and gently agitated to mix well. The cells in each well were left to stand at ambient temperature for 15 min, avoiding direct sunlight. The optical density was read at 570 nm with an ELISA reader (VersaMax, Molecular Devices, Sunnyvale, CA, USA). 

### 4.5. Oil Red O Staining

HepG2 cells were washed with PBS twice, fixed with 10% neutral-buffered formalin for 1 h and then stained with Oil Red O dye solution (Oil Red:H_2_O = 6:4) for 1 h at room temperature. The stained lipid droplets in cells were examined and photographed under a phase-contrast inverted microscope (Olympus, Tokyo, Japan) at 200× magnification. The stained lipids in cells were extracted using 100% isopropanol and quantified spectrophotometrically at 500 nm with an ELISA reader [41]. 

### 4.6. Nile Red Staining

The method of Phillip et al. was followed. Briefly, the medium was removed and the cells were rinsed twice to thrice with PBS. Fresh medium containing 10 μM Nile red was replaced and incubated for 10 min. The dye-containing medium was removed and the residual dye-containing medium was rinsed with PBS. The cells were treated with trypsin–EDTA and the cells were collected by centrifuging at 1250× *g* for 5 min. The trypsin–EDTA was removed and PBS was added, the cells were dispersed and the intensity of fluorescence was measured with a flow cytometer (Coulter Epics XL, Beckman, Ramsey, MN, USA). At the same time, the image was taken with the Olympus (IX71, Tokyo, Japan) inverted fluorescence microscope [42].

### 4.7. Assay for Intracellular ROS Content 

The protocol of Crow et al. was followed. In brief, the medium was removed, and the cells were rinsed twice to thrice with PBS. Fresh medium containing DCFH-DA (2′,7′-dichlorodihydrofluorescin diacetate) was introduced and the cells were incubated in a cell incubator for 30 min. The cell dye was removed, and the residual dye-containing medium was removed by rinsing with PBS. The cells were harvested with the aid of trypsin–EDTA. The collected cells were centrifuged at 1250× *g* for 5 min and the trypsin–EDTA was removed. The cells were dispersed with the aid of PBS and subjected to flow cytometric analysis to measure the intracellular fluorescence intensity. At the same time, the image was taken with the inverted fluorescence microscope [43].

### 4.8. Assay for Protein Content 

According to the method of Bradford et al., the cells were harvested with the aid of trypsin–EDTA and centrifuged at 1250× *g* for 5 min to remove trypsin–EDTA. The cells were dispersed in PBS, disrupted by ultrasonication and centrifuged at 13,000× *g* for 5 min to obtain the supernatant. To 5 μL of this, 795 μL deionized water was added, and after being well-agitated, 200 μL of protein assay dye was added, mixed well and left to stand for 10 min to facilitate reaction. The optical density of the reaction mixture (200 μL) was read at 595 nm with the ELISA reader. A calibration curve was established using bovine serum albumin (BSA) as the authentic sample [44]. 

### 4.9. Assay for Malondialdehyde Content 

Cells were harvested with the aid of trypsin–EDTA and centrifuged at 1250× *g* for 5 min. Trypsin–EDTA was removed. The cells were rinsed and dispersed in PBS, ultrasonicated and centrifuged at 13,000× *g* for 5 min to obtain the supernatant, which was subjected to MDA analysis with an ab118970 Lipid Peroxidation (MDA) assay kit (Abcam, Cambrensis, UK). Briefly, to 10 μL of the supernatant, 300 μL MDA lysis buffer was added and mixed well. BHT (100×, 3 μL) was added and centrifuged at 13,000× *g* for 10 min to obtain the supernatant. To 200 μL of the supernatant, 600 μL TBA was added, heated at 95 °C for 60 min and left to stand to cool to room temperature. A volume of 200 μL of the reaction mixture was depicted and the optical density was read at 532 nm with an ELISA reader. A calibration curve was established using the MDA standard attached in the kit [45].

### 4.10. Assay for the Intracellular Antioxidative Enzymes

#### 4.10.1. Assay for Superoxide Dismutase (SOD) Activity

As instructed by Marklund et al., the cells were harvested with the aid of trypsin–EDTA, and centrifuged at 1250× *g* for 5 min to remove trypsin–EDTA. The cells were rinsed with and dispersed in PBS, ultrasonicated and centrifuged at 13,000× *g* for 5 min to obtain the cytoplasm-containing supernatant. To 10 μL of the supernatant, 1 mL Tris-HCl (50 mM, containing 0.1 mM EDTA) and 5 μL pyrogallol (50 mM) were added. The optical density was read at 325 nm. The change in OD was taken every 30 s within a time span of 3 min [46]. The intracellular activity of SOD was calculated and expressed as U/mg protein. 

#### 4.10.2. Assay for the Catalase Activity

The method of Cohen et al. [47] was followed. In brief, the cells were harvested with the aid of trypsin–EDTA, and centrifuged at 1250× *g* for 5 min. The trypsin–EDTA was discarded and the cells were dispersed in PBS, ultrasonicated and centrifuged at 13,000× *g* for 5 min to obtain cytoplasm-containing supernatant. To 10 μL of this, 1 L of Tris-HCl (50 mM, containing 0.1 mM EDTA) and 900 μL H_2_O_2_ (10 mM) were added. The OD was taken at 240 nm, and the change in OD was read every 30 s within the first 3 min [47]. The intracellular catalase activity was calculated and expressed as μmol H_2_O_2_/min/mg protein. 

#### 4.10.3. Assay for the Glutathione Peroxidase Activity

As instructed by Habig et al., the cells were harvested with the aid of trypsin–EDTA and centrifuged at 1250× *g* for 5 min. The trypsin–EDTA was discarded, and the cells were dispersed in PBS, ultrasonicated and centrifuged at 13,000× *g* for 5 min to obtain the cytoplasm-containing supernatant. To 10 μL of this, 90 μL PBS (0.1M), 880 μL GSH (1 mM) and 20 μL CDNB (50 mM) were added and mixed well. The OD at 340 nm was taken, and the change in OD was read every 30 s within the initial 3 min time span [48]. The glutathione peroxidase activity was calculated and expressed as nmol CDNB-GSH nmol/mg protein. 

### 4.11. Assay for the Inflammatory Factors

#### 4.11.1. Assay for the TNF-α Content

The expressed content of TNF-α was measured by following the protocol as instructed by BosterBio (Pleasanton, CA, USA) using the Human TNF Alpha PicoKine™ ELISA Kit (BosterBio, Pleasanton, CA, USA). Upon addition of the final 90 μL TMB color- developing agent, the plate hole was closed and the cells were incubated at 37 °C for 20 min. To each hole, 100 μL TMB stop solution was introduced and the OD was read at 450 nm within 30 min using the ELISA reader.

#### 4.11.2. Assay for the IL-6 Content

A Human IL-6 PicoKine™ ELISA Kit (BosterBio, Pleasanton, CA, USA) was used to determine the IL-6 content following the instructions provided by BosterBio (Pleasanton, CA, USA). Upon addition of the final 90 μL TMB color-developing agent, the plate hole was closed and the cells were incubated at 37 °C for 20 min. To each hole, 100 μL TMB stop solution was introduced and the OD was read at 450 nm within 30 min using the ELISA reader. 

#### 4.11.3. Assay for the IL-1β Content

Human IL-1 Beta PicoKine™ ELISA Kit (BosterBio, Pleasanton, CA, USA) was used to determine the IL-1β content by following the instructions provided by BosterBio (Pleasanton, CA, USA). Upon addition of the final 90 μL TMB color-developing agent, the plate hole was closed and the cells were incubated at 37 °C for 20 min. To each hole, 100 μL TMB stop solution was introduced and the OD was read at 450 nm within 30 min using the ELISA reader.

### 4.12. Analysis for Gene Expression

#### 4.12.1. Extraction of RNA

Following the instructions provided by Gene Mark (Atlanta, GA, USA), the total RNA was extracted using the total RNA purification kit (TR01-150) (Gene Mark, Atlanta, GA, USA). The RNA obtained was freeze-stored at −80 °C.

#### 4.12.2. Reverse Transcription to Obtain cDNA

The purity of RNA was assessed by the microspectrophotometer CB-4500 (CLUBIO, Taiwan). The content of RNA was fixed at 3 μg and processed with PrimeScriptTM Reagent Kit (TaKaRa). A total of 2 μL 5X-PrimeScript Buffer, 0.5 μL PrimeScript RT Enzyme Mix1, 0.5 μL Oligo dT primer (50 μM), 0.5 μL Random 6 mer (100 μM) and 3 μL RNA were introduced and mixed well. The total volume was made to equal 10 μL with the RNase-free dH_2_O. the mixture was incubated at 37 °C for 2 h. The cDNA was formed from the reverse transcription of RNA. The cDNA obtained was stored at −80 °C for use (cDNA Template). 

#### 4.12.3. Real-Time Polymerase Chain Reaction 

KAPA SYBR^®^ Fast Abi Prism^®^ (KAPA Biosystems) was used, to which 10 μL 2X-KAPA SYBR FAST qPCR Master Mix, 0.4 μL of 10 μM primer (Forward), 0.4 μL of 10 μM primer (Reverse) and 1 μL of cDNA Template (×10 dilution) were introduced. Finally, the volume was made to equal 20 μL using either RNase-free water or sterilized deionized water. The mixture was placed into StepOnePlusTM Real-Time PCR System (Applied Biosystems, Thermo Fisher). The condition was set as: preheating at 95 °C for 10 min to activate the enzymes, then entering the cycle of 95 °C, 15 s denaturation, 60 °C 60 s. A total 40 cycles were carried out. On finishing, the 2^−ΔΔCT^ value for each sample was calculated. The primer pairs for RT-PCR used in the study are presented in Appendix A.

### 4.13. SDS-PAGE

#### 4.13.1. Separation of Proteins by Electrophoresis 

The resolving gel (the lower gel) was prepared by mixing 6 mL H_2_O, 1.3 mL Tris-HCl (pH 8.8), 2 mL acrylamide (30%), 50 μL SDS (10%), 50 μL APS (10%) and 5 μL TEMED. The stacking gel (the upper gel) was made from 1.4 mL H_2_O, 205 μL Tris-HCl (pH 6.8), 330 μL acrylamide (30%), 20 μL SDS (10%), 20 μL APS (10%) and 2 μL TEMED. The two gels were prepared following Merck technological guidance. 

The finished solidified gels were inserted into the electrophoretic chamber. A 1× running buffer was added (1 part of 10× dilution from the stocking running buffer, which is prepared from 10× running buffer containing 30.2 g Tris-base, 144 g glycine, 10 g SDS and 1L water; this stock is diluted 10 fold before use) to reach the height of solvent required by the chamber. On the other hand, the protein samples were diluted with 5× running buffer in a 4:1 ratio, mixed well and heated at 95 °C for 5 min, centrifuged at 16,000× *g* for 5 min, left to cool and pipetted into the sample wells. One of the wells was introduced with 2 μL pre-stain protein marker. The electrophoresis was started at 90 V for 120 min [20]. 

#### 4.13.2. Electrotransfer

The polyvinylidene-difluoride (PVDF) membrane (thickness 0.2 μm) was cut into a suitable size and dipped into 100% methanol. Similarly, the filter papers required for the stacking and resolving layers were cut into suitable sizes and dipped into the transfer buffer (prepared by mixing 6 g Tris-base, 2.9 g glycine, 0.37 g SDS, 200 mL methanol and 800 mL water). The semi-dry transfer method was followed. In brief, the dipped filter papers were placed onto the slat mold of the transfer and pressed to expel the air bubbles, then topped with the methanol-dipped PVDF membrane, on which the dipped filter papers were placed and gently pressed to remove the air bubbles. The transfer cover was closed and a 0.25 A current was applied to process the transfer for 45 min [20]. 

#### 4.13.3. Western Blot

After transfer, the PVDF membrane was blocked with 5% skimmed milk (prepared by incorporation of 5% skimmed milk into 1×TBST buffer which was a 10-fold diluted buffer from the 10×-TBS buffer (prepared by mixing 40 g NaCl, and 12.1 g Tris-base and 500 mL water; after dilution, 0.1% Tween-20 was added before use). The blocking took 24 h at 4 °C with constant agitation. The blocking skimmed milk was decanted and rinsed with TBST. The suitably dispensed primary antibody (in TBST) was applied and agitated at 4 °C for 16 h. The primary antibody was removed and the PVDF membrane was rinsed thrice with TBST, each time for 10 min. The secondary antibodies prepared in an appropriate ratio in TBST were applied and agitated at 4 °C for 30 to 60 min. The required reaction time depends on the sensitivity of the antibodies. The secondary antibodies were removed and the PVDF membranes were rinsed thrice with TBST, each time for 10 min. The solutions A and B of the Immobilon Western Chemiluminescent HRP Substrate (WBKLS0500, Millipore) were mixed at a 1:1 ratio. TBST was used to adjust the appropriate concentration of the antibodies if necessary, depending on the sensitivity of the antibodies. The finished PVDF membranes were dipped into the above solution and the image was taken with the imaging system. The Image J Processing System was used to compare the intensity of fluorescence [20].

### 4.14. Statistical Analysis

Statistical analyses were performed using SigmaPlot v15.0 (Systat *Software* Inc., San Jose, CA, USA). Statistical comparisons between study groups were analyzed by Student’s t-test. *p*-values < 0.05 were considered statistically significant. 

## 5. Conclusions

The present study reinforces the inhibitory effects of styrylpyrone-enriched extract—SPEE—prepared from PL mycelia on hepatic steatosis and offers some important insights into the mechanisms that may underlie these activities. The current study provides knowledge of the novel mechanism by which SPEE inhibits palmitate-induced oxidative stress through the activation of the protein expressions of SIRT1, AMPK and PGC1-α and suppression of NFκB to inhibit the production of P/O-induced pro-inflammatory factors and cell apoptosis. Further studies conducted in animals and followed by clinical studies to provide the critically important experimental data may unveil a potential therapeutic application of SPEE for treating NAFLD.

## Figures and Tables

**Figure 1 ijms-24-03672-f001:**
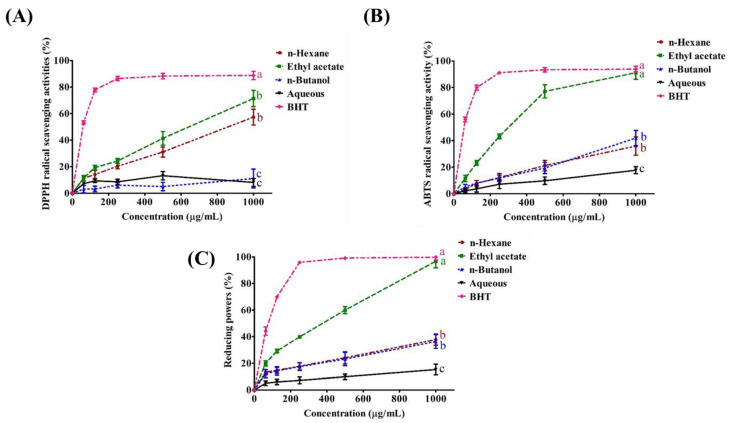
The antioxidant activities of 75% methanol crude extract and its different solvent partitions including n-hexane, ethyl acetate, n-butanol and water from freeze-dried powder of *Phellinus linteus* mycelia. Each value represents the mean ± standard deviation (SD, n = 3) SD of triplicate experiments. (**A**) DPPH radical scavenging activity, (**B**) ABTS radical scavenging activity and (**C**) reducing power on ferric ions. Means with different letters are significantly different (*p* < 0.05).

**Figure 2 ijms-24-03672-f002:**
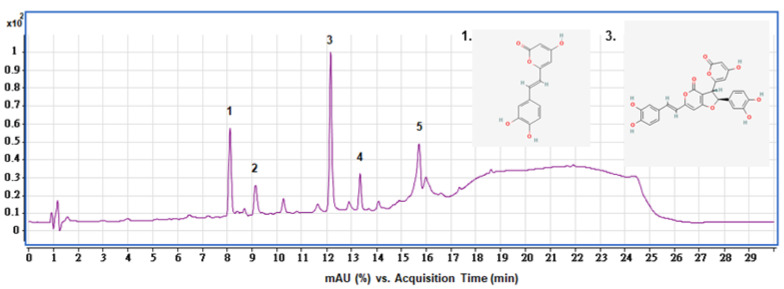
Representative HPLC profile of main constituents present in the SPEE. The diode-array detector was utilized at the wavelength of 210–600 nm in the HPLC analysis. Peak 1: hispidin; 2: MW. 490; 3: hypholomine B; 4: hypholomine B isomer; 5: unidentified compound. Chemical structures of 1 and 3 from PubChem are cited in the figure.

**Figure 3 ijms-24-03672-f003:**
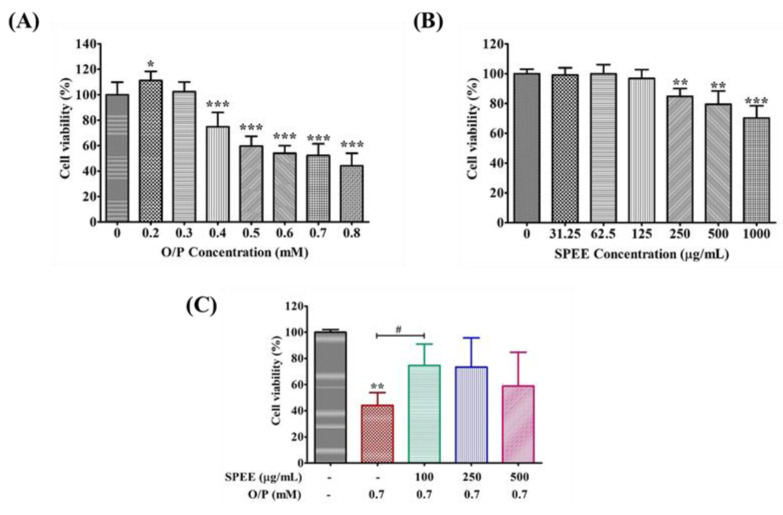
The effects of O/P SPEE and their combined treatments on HepG2 cell proliferation at 24 hr. (**A**) The viability of HepG2 cells. (**B**) The cytotoxicity of SPEE on HepG2 cells. (**C**) The rescue of HepG2 cells treated with O/P 0.7 mM by SPEE within a dose range of 100 to 500 μg/mL. Data are expressed as mean ± SD from triplicate experiments (*n* = 3). The Student’s t-test was used to compare the means between two groups. * *p* < 0.05, ** *p* < 0.01, *** *p* < 0.001 vs. the control group. ^#^ *p* < 0.05 vs. the O/P group.

**Figure 4 ijms-24-03672-f004:**
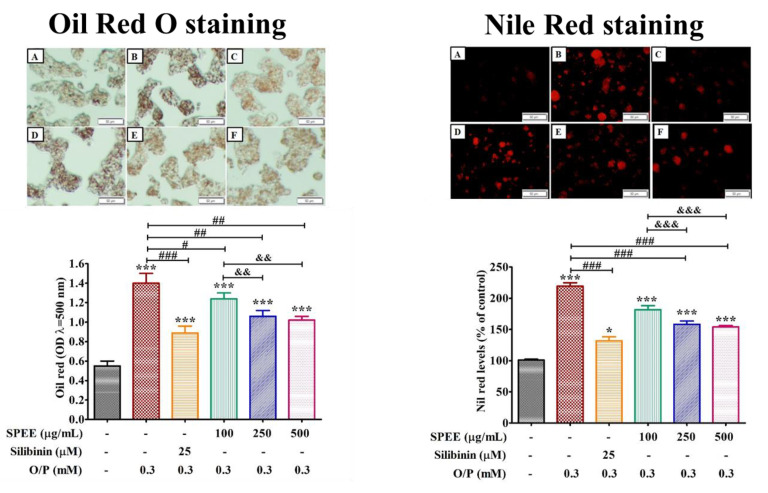
The effects SPEE on O/P-induced intracellular lipid accumulation analyzed by Oil Red O staining and Nile red staining in HepG2 cell. HepG2 cells were co-treated with 0.3 mM O/P and SPEE at 100, 150 and 500 μg/mL, respectively, for 24 h. Silibinin (25 μM) was used as the positive control. Oil-Red-stained cells were observed under inverted microscope and Nile-red-stained cells were assessed by phase-contrast fluorescence microscopy (magnification, ×200). (**A**) Control, (**B**) O/P 0.3 mM, (**C**) O/P 0.3 mM + Silibinin 25 µM, (**D**) O/P 0.3 mM + SPEE 100 μg/mL, (**E**) O/P 0.3 mM + SPEE 250 μg/mL and (**F**) O/P 0.3 mM + SPEE 500 μg/mL. Values are expressed as the mean ± SD (*n* = 3). * *p* < 0.05 and *** *p* < 0.001 vs. the control group. ^#^ *p* < 0.05, ^##^ *p* < 0.01 and ^###^ *p* < 0.001 vs. the O/P group. ^&&^ *p* < 0.01 and ^&&&^ *p* < 0.001 vs. the O/P 0.3 mM + SPEE 100 μg/mL group.

**Figure 5 ijms-24-03672-f005:**
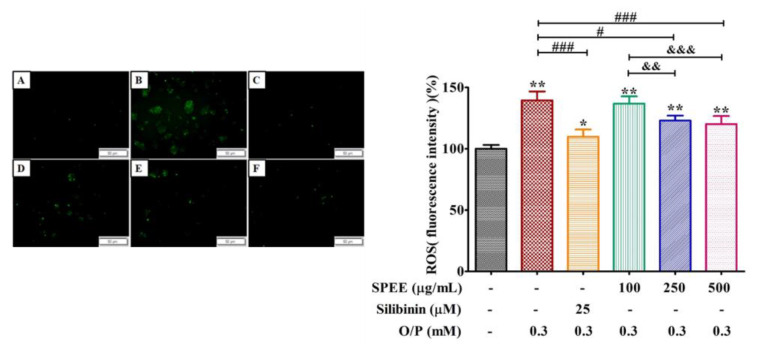
The effects SPEE on O/P-induced intracellular reactive oxygen species (ROS). HepG2 cells were co-treated with 0.3 mM O/P and SPEE at 100, 150 and 500 μg/mL, respectively, for 24 h. Silibinin (25 μM) was used as the positive control. Intracellular ROS were assessed using DCFH-DA fluorescence staining and were observed by phase-contrast fluorescence microscopy (left panel, original magnification ×100). (**A**) Control, (**B)** O/P 0.3 mM, (**C**) O/P 0.3 mM + Silibinin 25 µM, (**D**) O/P 0.3 mM + SPEE 100 μg/mL, (**E**) O/P 0.3 mM + SPEE 250 μg/mL and (**F**) O/P 0.3 mM + SPEE 500 μg/mL. Values are expressed as the mean ± SD (*n* = 3, right panel). * *p* < 0.05 and ** *p* < 0.01 vs. the control group. ^#^ *p* < 0.05 and ^###^ *p* < 0.001 vs. O/P group. ^&&^ *p* < 0.01 and ^&&&^ *p* < 0.001 vs. O/P 0.3 mM + SPEE 100 μg/mL group.

**Figure 6 ijms-24-03672-f006:**
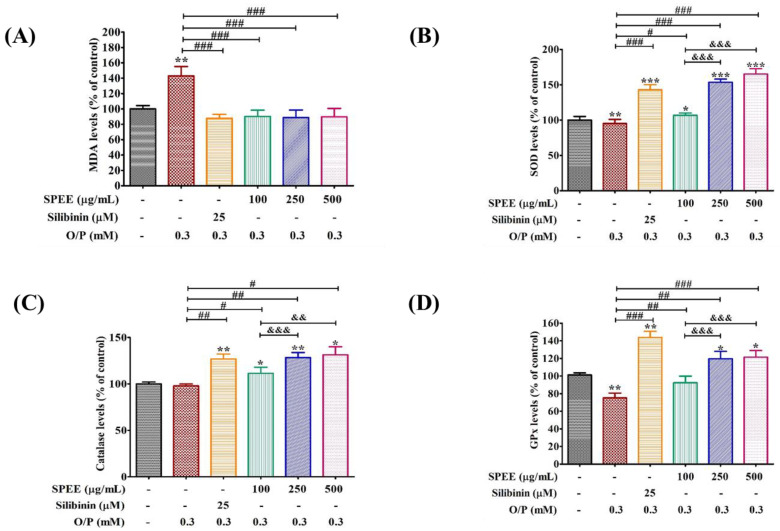
The effects of SPEE on O/P-inhibited antioxidant activities in HepG2 cells. HepG2 cells were co-treated with 0.3 mM O/P and SPEE at 100, 250 and 500 μg/mL for 24 h. SPEE inhibited the O/P-induced MDA production (**A**), and increased the antioxidant enzyme activities of SOD (**B**), catalase (**C**) and GPx (**D**). Silibinin (25 μM) was used as the positive control. Values are expressed as mean ± SD from triplicated experiments (*n* = 3). * *p* < 0.05, ** *p* < 0.01 and *** *p* < 0.001 vs. the control group. ^#^ *p* < 0.05, ^##^ *p* < 0.01 and ^###^ *p* < 0.001 vs. the O/P group. ^&&^ *p* < 0.01 and ^&&&^ *p* < 0.001 vs. the O/P 0.3 mM + SPEE 100 μg/mL group.

**Figure 7 ijms-24-03672-f007:**
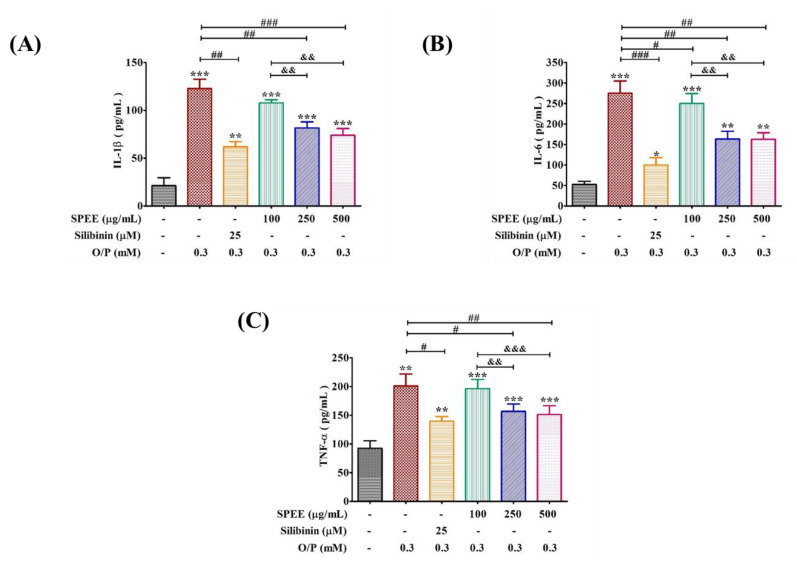
The effects of SPEE on O/P-induced intracellular inflammatory factors in HepG2 cells. HepG2 cell lines were co-treated with 0.3 mM O/P and SPEE (100, 250 and 500 μg/mL, respectively) for 24 h. The expressions of pro-inflammatory factors IL-1β (**A**), IL-6 (**B**) and TNF-α (**C**) were analyzed as described in Materials and Methods. Silibinin was used as the positive control. Data are expressed as mean ± SD from triplicate experiments. * *p* < 0.05, ** *p* < 0.01 and *** *p* < 0.001 vs. the control group. ^#^ *p* < 0.05, ^##^ *p* < 0.01 and ^###^ *p* < 0.001 vs. the O/P group. ^&&^ *p* < 0.01 and ^&&&^ *p* < 0.001 vs. the O/P 0.3 mM + SPEE 100 μg/mL group.

**Figure 8 ijms-24-03672-f008:**
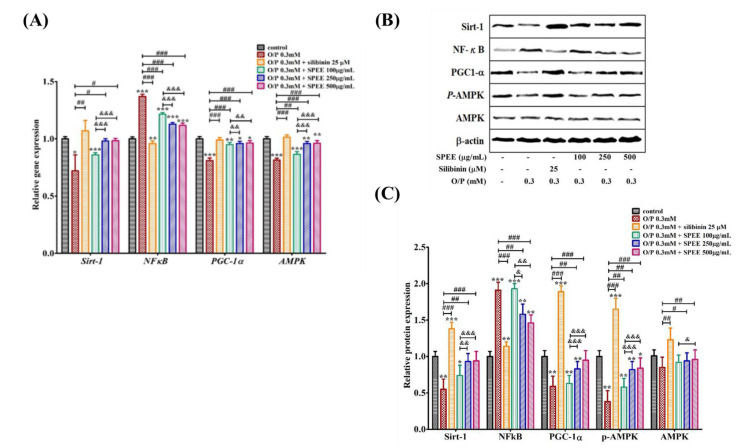
The effects of SPEE on O/P-induced m-RNA (**A**) and protein expressions (**B**,**C**) in HepG2 cells. HepG2 cell lines were co-treated with 0.3 mM O/P and SPEE (100, 250 and 500 μg/mL, respectively) for 24 h. Silibinin was used as the positive control. Data are expressed as mean ± SD from triplicate experiments. * *p* < 0.05, ** *p* < 0.01 and *** *p* < 0.001 vs. the control group. ^#^ *p* < 0.05, ^##^ *p* < 0.01 and ^###^ *p* < 0.001 vs. the O/P group. ^&^ *p* < 0.05, ^&&^ *p* < 0.01 and ^&&&^ *p* < 0.001 vs. the O/P 0.3 mM + SPEE 100 μg/mL group.

**Figure 9 ijms-24-03672-f009:**
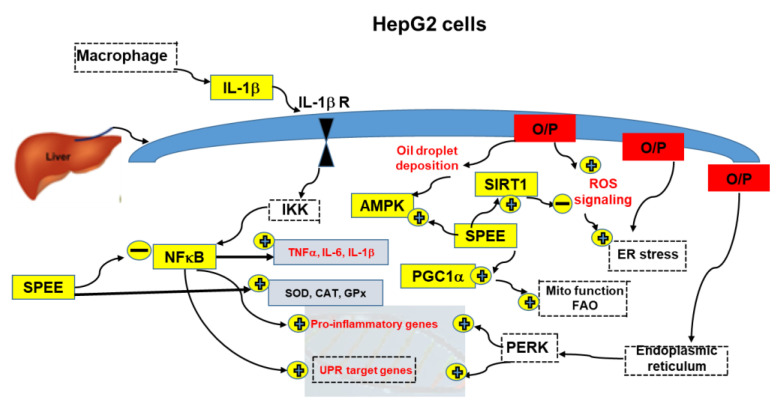
O/P insults HepG2 cell to induce ROS signaling via ER stress associated with oil droplet deposition, which enhances energy stress and upregulates AMPK, Sirt1 and, subsequently, PGC1-α. SPEE further upregulates AMPK, Sirt1 and PGC1-α. Simultaneously, O/P stimulates macrophage to produce IL-1β, which activates NFκB to upregulate antioxidant and pro-inflammatory cytokines TNF-α, IL-6 and IL-1β. On the other hand, O/P may upregulate PERK which in turn upregulates the pro-inflammatory genes and UPR target genes. SPEE downregulated pro-inflammatory cytokines; conversely, it upregulated the antioxidant enzymes’ activities. Finally, the upregulated PGC1-α improved the mitochondrial function (Mito function) and fatty acid oxidation (FAO). Items with dotted lines and dotted boxes are depicted from the literature and were not included in this experiment.

## Data Availability

Data is contained within the article.

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
