# Peer review of "Inhibitory Effect of Styrylpyrone Extract of Phellinus linteus on Hepatic Steatosis in HepG2 Cells"

_ijms, 2023, doi:10.3390/ijms24043672_

Round 1

Reviewer 1 Report

The article presented is Interesting and scientifically solid, there are however a few minor adjustments.

1)The abstract showed be revisited and rewritten to be more concise as example the sentence above is more suitable for the introduction.

“In the previous study, the highest antioxidants extract enriched in styrylpyrone components (designated as SPEE) is obtained from the 75% methanol extraction followed by the ethyl acetate partition of the extract.”

2) Normalize de nomenclature for ABTS to ABTS•+

3)Line 31 define O/P

4)Line 36 define AMP .

5)They are too many keywords; II suggest removing antioxidant; anti-inflammatory;

 free fatty acid; and styrylpyrones;

6) in the introduction change ; for , .

7)Line 77 The reference is in the wrong style.

 8)Lines 88 (PL-Hx, 9.24 ± 0.04 mg/g), ethyl acetate (SPEE, 52.58 ± 0.03 mg/g), n-butanol 88 (PL-BN, 11.11 ± 0.03 mg/g), and water (PL-W) (1.87 ± 0.04 mg/g)

Abbreviations have to show consistency and followed the same logic. So SPEE should be PL-EA Fin alternative don’t use abbreviations for the other fractions.

 9) It Is necessary to rewrite all titles in the manuscript.

Is not acceptable to include results and results discussion in titles.

As An exemple there are a few cases:

2.2 SPEE exhibited strong antioxidant activities

2.3. HPLC analysis indicated hypholomine B and hispidin to be the main constituents in SPEE

3.3 SPEE dose-dependently inhibited the intracellular lipid droplet deposition caused a severe Oxidative stress followed by the potential ER stress

 10) It Is necessary to rewrite all figure legends in the manuscript. They are too long and include unnecessary information as example for figure 3.

 11) “The effects of oleic and palmitic acid (O/P), ethyl acetate fraction from 75% methanol extract SPEE of Phellinus linteus mycelium freeze-dried powder and their combined treatments on HepG2 cell proliferation at 24 hr. A) The viability of HepG2 cells was inhibited in a dose dependent manner by O/P ranging from 0.2 to 0.8 mM. B) The cytotoxicity of SPEE on HepG2 cells revealed to be in a dose dependent fashion within 63 to 1000.g/mL. C) The rescue of HepG2 cells insulted with O/P 0.7 mM by SPEE within a dose range 100 to 500 .g/mL. Data are expressed as mean .SD from triplicate experiments (n = 3). The Student's t test was used to compare the means between two groups. *P < 0.05, ** P < 0.01, *** P < 0.001 versus the control group.#P < 0.05 versus the O/P group.”

 Acronymous were already defined REMOVE ethyl acetate fraction from 75% methanol extract (SPEE) and oleic and palmitic acid (O/P). use the abbreviation.  cells was inhibited in a dose dependent manner by O/P ranging from 0.2 to 0.8 mM. this is a results presentation it should not be included in the figure legend.

 12) Line 141define DCFH-DA

 13) SPEE rescued HepG2 cells from insults by O/P

In the context of biology, the term "insult" typically refers to an injury or damage to an organism's tissues or cells. But the most common interpretation of the word is to be treat or speak to insolently or with contemptuous rudeness; affront. For that, I advise using a synonymous.

 14) Provide references missing in the following sections of Material and methods: Induction of HepG2 cell line with O/P; Culture of HepG2 Cells; Assay for malondialdehyde content ;SDS-PAGE; Electrotransfer; Western Blot

Author Response

Please read the attached file. 

Reviewer 2 Report

Non-alcoholic fatty liver disease (NAFLD) is a clear risk for patients and it is scientifically advisable to explore available inhibitors from natural products, especially active components of medicinal fungi (mushrooms). styrylpyrone analogues are characteristic active components of mulberry and "mulberry-like fungi, which exhibit significant antioxidant activity. This manuscript is the first to investigate the curative effect of styrylpyrone analogues on NAFLD. The molecular mechanism of the anti-NAFLD activity of styrylpyrone analogues is explained by measuring physiological and biochemical parameters and validating the signaling pathway by molecular biology. The experimental design is sound, the results are reliable, and the TOPIC of the manuscript is in line with the SCOPE of the journal.

However, before accepting, several bugs or concerns need to be addressed. For details, see the following comments.

1.      The format of insertion and citation of the literature does not conform to the format prescribed by IJMS, please correct.

2.      In Figure 2, the structures of the four known compounds should be shown.

3.      In line 646,648,651, Species names in Latin need to be italicized.

4.      In line 431,1250´gshould be 1250 ´ g.

5.      In Figure 2, the detection wavelength of the HPLC should be indicated.

6.      In Figure 4 and 5, the subfigure labels should be A, B, not I, II.

7.      In Figure 4, the subfigure labels should be A, B, not I, II.

8.      The third paragraph of the Introduction section, which should focus on the structural diversity and activity diversity of the styrylpyrone compounds, cites the following literatures, please.

A.     Styrylpyrone-class compounds from medicinal fungi Phellinus and Inonotus spp., and their medicinal importance, 10.1038/ja.2011.2.

B.     Diverse Metabolites and Pharmacological Effects from the Basidiomycetes Inonotus hispidus, 10.3390/antibiotics11081097.

9.      The supplementary material listed in the manuscript is not consistent with the supplementary material provided on the website.

Author Response

Please read the attached file. 
